# Association Between Systemic Neuroinflammation, Pain Perception and Clinical Status in Fibromyalgia Patients: Cross-Sectional Study

**DOI:** 10.3390/cells13201719

**Published:** 2024-10-17

**Authors:** María Elena González-Álvarez, Víctor Riquelme-Aguado, Ángela González-Pérez, Rosa Murillo-Llergo, María Manjón-Olmedillas, Silvia Turroni, Giacomo Rossettini, Jorge Hugo Villafañe

**Affiliations:** 1Escuela Internacional de Doctorado, Rey Juan Carlos University, 28008 Madrid, Spain; 2Department of Physical Therapy, Occupational Therapy, Rehabilitation and Physical Medicine, Rey Juan Carlos University, 28032 Madrid, Spain; 3Cognitive Neuroscience, Pain, and Rehabilitation Research Group (NECODOR), 28032 Madrid, Spain; 4Department of Basic Health Sciences, Rey Juan Carlos University, 28933 Madrid, Spain; victor.riquelme@urjc.es; 5Grupo de Investigación Consolidado de Bases Anatómicas, Moleculares y del Desarrollo Humano de la Universidad Rey Juan Carlos (GAMDES), 28922 Alcorcón, Spain; 6Fisioterapia Oreka CB, 45200 Illescas, Spain; 7Laboratorio de Análisis Clínicos de la Dra. González Pérez, 28020 Madrid, Spain; informacion@laboratorioangelagonzalez.com (Á.G.-P.); ros-muri@hotmail.com (R.M.-L.); maria.manjo.olmedillas@gmail.com (M.M.-O.); 8Unit of Microbiome Science and Biotechnology, Department of Pharmacy and Biotechnology, University of Bologna, Via Belmeloro 6, 40126 Bologna, Italy; silvia.turroni@unibo.it; 9Department of Physiotherapy, Faculty of Sport Sciences, Universidad Europea de Madrid, 28670 Villaviciosa de Odón, Spain; mail@villafane.it; 10Musculoskeletal Pain and Motor Control Research Group, Faculty of Sport Sciences, Universidad Europea de Madrid, 28670 Villaviciosa de Odón, Spain

**Keywords:** fibromyalgia, interleukins, neuroinflammation

## Abstract

Background: Fibromyalgia (FM) is characterized by chronic pain and a complex array of symptoms, with neuroinflammation implicated in its pathophysiology. Methods: This study aimed to explore the association between neuroinflammation, measured through interleukin levels (IL-1, IL-6, IL-8), and clinical outcomes in FM patients. Using a cross-sectional study design, blood levels of these interleukins were correlated with pain severity and disability, assessed via the Fibromyalgia Impact Questionnaire (FIQ) and pain measures. Results: Results indicated that IL-6 and IL-8 may particularly serve as biomarkers for pain severity and disability in FM patients, showing significant associations with worse clinical outcomes. Elevated IL-8 levels, for instance, correlated strongly with increased pain perception and higher disability scores. Conclusions: These findings suggest that specific interleukins are not only elevated in FM but are actively involved in the modulation of pain and disability, underscoring the role of systemic neuroinflammation in the clinical severity of FM. This study contributes to a deeper understanding of the inflammatory mechanisms in FM and underscores the potential of targeting interleukins in therapeutic strategies.

## 1. Introduction

Fibromyalgia (FM) is a complex disorder with multiple dimensions, where pain is the main symptom. This condition also involves other notable physical, psychological, and cognitive issues, such as fatigue, poor-quality sleep, symptoms of depression and anxiety, and cognitive distortions like pain catastrophizing [1,2]. In Madrid, Spain, the prevalence of FM is around 5% of the population, with women aged 46 to 60 being the most affected. Despite considerable research efforts, the cause of FM is still unclear, making effective clinical management a challenge [3,4].

Inflammatory substances contribute to the regulation of pain by disrupting nociceptive transduction, conduction, and transmission. This regulation may occur through changes in the transcription rate and/or posttranslational modifications of proteins associated with the pain pathway [5,6]. Neuroinflammation entails immune cell activation in the central nervous system, contributing to chronic pain. FM is a complex, actively researched area involving chronic glial cell activation [7], neuronal plasticity changes [8], and interactions with psychological and environmental factors [9].

Interleukins (ILs) play a crucial role in modulating pain through their influence on the inflammatory response and the nervous system. Pro-inflammatory ILs like IL-1β and IL-6 increase pain sensitivity by promoting inflammation and contributing to both central and peripheral sensitization. Moreover, the presence of pro-inflammatory ILs in chronic pain conditions can lead to the overexpression of N-methyl-d-aspartate (NMDA)- and α-amino-3-hydroxy-5-methyl-4-isoxazolepropionic acid (AMPA) receptors, which are associated with the pathophysiology of nociplastic pain. This mechanism is essential for long-term potentiation, a critical process for synaptic plasticity and memory [10,11]. This heightened sensitivity can amplify pain perception, even from non-painful stimuli. In chronic pain conditions like rheumatoid arthritis and FM, persistent pro-inflammatory ILs sustain inflammation and pain. In contrast, IL-10 has an anti-inflammatory function and is considered a protective factor against chronic pain. In FM patients, its presence is decreased, which may play an important role in the malfunctioning of the immune system in these patients [12]. Thus, the balance of ILs significantly impacts pain perception and chronicity. However, ILs that have been studied in FM have yielded conflicting evidence. For example, some studies have shown elevated levels of IL-1β [13], while others have found no differences [14]. Regarding IL-6, higher serum levels were found in FM patients and were significantly associated with fatigue and pain ratings [15]. Elevated IL-6 levels in the peripheral blood of FM patients were linked to hyperalgesia [16]. IL-6 released by thalamic mast cells contributes to inflammation and pain and may stimulate thalamic nociceptive neurons [17]. However, another study [18] found no differences after adjusting for age as a covariate. In some previous studies, pain severity has been positively correlated with IL-8 levels [19,20], although one study [14] found no differences compared to healthy subjects. Variations also exist in the sampling protocols and analysis methods (for serum, skin biopsies, muscle samples, cerebrospinal fluid, and intracellular levels of mononuclear blood cells).

Interleukins are not the only biomarker that has been found in patients with chronic pain. An altered postprandial glycemic response in chronic pain patients has also been previously shown [21]. It serves as a key indicator of carbohydrate metabolism and impacts multiple facets of metabolic health. Altered glucose metabolism may reduce the ability to manage postprandial glucose spikes, potentially leading to energy fluctuations and increased systemic inflammation [22]. In particular, in relation to IL-6, it has been suggested that “oscillatory postprandial glycemic fluctuations are particularly detrimental” [23]. High levels of IL-6 have been linked to insulin resistance and can impact postprandial blood glucose levels. In chronic inflammatory conditions like obesity, elevated IL-6 levels may contribute to the impaired regulation of post-meal glucose [24]. IL-1 may also play a role in postprandial glucose regulation. IL-1β has been implicated in the dysfunction of insulin-producing pancreatic beta cells and in insulin resistance [25].

Previous scientific literature on the role of ILs in FM is limited. Further research is needed to delve into the clinical implications of ILs in these patients. We hypothesized that elevated levels of IL-1, IL-6, and IL-8 might be associated with worse clinical status and impaired pain inhibitory function in FM patients. Therefore, the aim of this study was to explore the potential association between neuroinflammation and clinical status in FM patients.

## 2. Materials and Methods

### 2.1. Study Design and Setting

This is a cross-sectional study of a randomized control trial approved by the ethics committee from the King Juan Carlos University number 1601202303523 and reported according to the “STROBE” checklist for observational studies [26]. The study was conducted in the Community of Madrid, Spain between October and November 2023. All participants signed an informed consent after receiving oral and written information before participating in the study. The main protocol is registered in ClinicalTrials.gov, with ID protocol number NCT05932433 in June 2023.

### 2.2. Participants

Only female participants were recruited for this study. Patients were included if they had been diagnosed with FM at least 6 months ago, were over 18 years old, and were Spanish-speaking. Exclusion criteria were having or having had cancer, psychiatric disorders, or other major illnesses, such as Lyme disease, hepatitis, or irritable bowel syndrome. Participants were recruited from “AFIBROM”, fibromyalgia association recommendations and social networks, and from social networks. They were contacted by phone or email and informed about the study. A total of 70 patients showed interest in our study. However, 10 refused to participate after the entire research plan and 7 were rejected because of the exclusion criteria. After the screening, 53 patients were included in the study to evaluate the experimental pain measurements and questionnaires. From these 53 participants, 30 women were randomly selected for blood analysis.

### 2.3. Procedure

Blood draws and all IL analyses were performed at the Dr. Ángela González Laboratory (Madrid, Spain) and were taken in the morning while fasting. Pain-related supplementary tests were performed at the Active Recovery Clinic (Madrid, Spain). Questionnaires and demographic, anthropometric, and clinical data were filled in online on the day of the pain sensory tests. Data included: age, height, weight, years since diagnosis, pain today, and mean pain in the last week. The questionnaires were the Brief Pain Inventory (BPI), Fibromyalgia Impact Questionnaire (FIQ), Rolland Morris, STAI (AR and AE), and questions about medication use during the last week. The blood analysis and the sensory tests were conducted on different days, with no more than 4 days between them.

### 2.4. Outcome Measures

#### 2.4.1. Interleukins

The technique used for IL-1 and IL-8 was enzyme-linked immunosorbent assay, performed using a manual data-independent acquisition (DIA source technique). For IL-6, the immunoassay technique was carried out using the Siemens IMMULITE equipment (immunoassay method). The entire process of blood collection, the evaluation of ILs, and the results was carried out at the Dr Ángela González Laboratory (Madrid, Spain).

#### 2.4.2. Experimental Pain Measurements

Pain measurements were performed by a physiotherapist, in a private room following the scheme of Figure 1. Firstly, the pressure pain threshold (PPT) was assessed twice in the middle phalanx of the middle finger of the dominant hand, 30″ apart. Afterwards, for the conditioned pain modulation (CPM) paradigm, a cuff was inflated in the non-dominant arm, until a feeling of 7 out of 10 on a pain scale was reached. Finally, the PPT was assessed again twice in the same ubication as before, with 30″ between the two measurements. Secondly, the PPT and CPM paradigms were also measured in the middle third of the right trapezius on the dominant side in the same way as described before. Finally, participants were asked about their drug use in the last 24 h, last 3 days, and last week.

#### 2.4.3. Self-Reported Pain Questionnaires

##### The Brief Pain Inventory (BPI)

The BPI is a self-administered tool that has been validated [27] and is commonly used to evaluate how pain affects daily activities. It employs a numerical rating scale (NRS) from 0 to 10 to assess pain intensity (sensory dimension) and its impact on the patient’s life (reactive dimension). Additionally, it includes questions about pain relief, pain characteristics, and the patient’s perception of the cause of the pain. The validated Spanish version of the BPI was utilized [28,29]. The NRS of pain was categorized based on the following cutoffs: 1–5 as mild, 6 as moderate, and 7–10 as severe [30,31].

##### Fibromyalgia Impact Questionnaire (FIQ)

The Spanish version of the FIQ was used to measure the disability and physical impact of FM. Total scores range from 0 to 100, with higher scores indicating a lower quality of life. The FIQ has demonstrated good psychometric properties and an internal consistency of 0.93 in Spanish FM patients [29,32]. An improvement of at least 30% has been identified as the minimum for a positive response to treatment [33]. The cutoff scores for the FIQ were severe for ≤59 and mild or moderate for >59 [34].

##### Demographics, Anthropometrics and Medication

Participants were asked their age, height, and weight. From these measurements, the body mass index (BMI) was calculated using the formula [BMI = weight (in kg)/height^2^ (in m^2^)] and then categorized according to the following criteria: underweight—BMI < 18.5 kg/m^2^; normal weight—BMI ≥ 18.5 to 24.9 kg/m^2^; overweight—BMI ≥ 25 to 29.9 kg/m^2^; obesity—BMI ≥ 30 kg/m^2^ [35]. Patients were also asked how many years they had been living with FM.

Participants were asked about medication use in the last 24 h, last 3 days, and last week. Due to the different characteristics of the medications (anti-inflammatory, opioid, analgesic, etc.), they were just asked about the time since last use, not about the type of the medication or the active ingredient. Participants were advised not to alter their usual medication routine during the study.

### 2.5. Data Analysis

Statistical analyses were performed using SPSS software version 29.0 (SPSS Inc., Chicago, IL, USA). Descriptive statistics, including means and standard deviations (SD), were utilized to summarize the demographic, anthropometric, and clinical characteristics of the study population. Pearson correlation coefficients were employed to assess the associations between IL levels (IL-1, IL-6, and IL-8) and pain metrics, specifically the NRS for the last 24 h and the last week, as well as functional disability as measured by the FIQ.

To further delineate the relationship between IL levels and clinical outcomes, both univariate and multivariate linear regression analyses were performed. These analyses evaluated the predictive ability of each IL for pain severity and functional disability, while adjusting for potential confounding variables such as age and BMI.

Participants were stratified into subgroups according to their IL levels to investigate variations in clinical outcomes across different inflammatory profiles. The interleukin levels were categorized based on the laboratory’s internal validation, as there are no universally standardized cut-offs for these interleukins across all clinical or research settings. For IL-1, the levels were stratified as follows: low levels ≤ 0.4 pg/mL, intermediate levels: 0.4 pg/mL–13.6 pg/mL, and high levels ≥ 13.6 pg/mL; for IL-6, low levels ≤ 2 pg/mL, intermediate levels: 2 pg/mL–5.9 pg/mL, and high levels ≥ 5.9 pg/mL; and for IL-8, low levels ≤ 20 pg/mL, intermediate levels: 20 pg/mL–50 pg/mL, and high levels ≥ 50 pg/mL. These stratifications follow the laboratory’s guidelines, although it is important to note that ranges may vary across different studies. Statistical significance for all tests was determined at a *p*-value threshold of <0.05

## 3. Results

### 3.1. Demographic, Anthropometric and Clinical Characteristics

Table 1 presents the demographic, anthropometric, and clinical characteristics of the study sample (*n* = 30). The study sample consisted exclusively of women with a mean age of 50.5 years (SD = 8.5), reflecting a middle-aged cohort. Participants had a mean height of 162.4 cm (SD = 6.44) and a mean weight of 72.8 kg (SD = 14.19), resulting in a mean BMI of 27.7 (SD = 5.88), indicating a prevalence of overweight individuals within the group. On average, participants had been experiencing FM symptoms for 10 years (SD = 8.70). Pain levels were reported as severe, with a mean NRS of 7 (SD = 1.74) over the past 24 h and 7.6 (SD = 1.83) over the past week. The mean FIQ score was 74.1 (SD = 15.43), indicating a significant and severe impact on daily functioning.

### 3.2. Interleukins vs. Pain and Disability

Participants were categorized based on their IL levels, as shown in Table 2. The majority (73.3%, *n* = 22) had IL-1 levels below 0.4 pg/mL, 16.6% (*n* = 5) had intermediate levels, and 10% (*n* = 3) had elevated IL-1 levels (>13.6 pg/mL). Regarding IL-6, 70% of participants (*n* = 21) had levels below 2 pg/mL, while the remaining 30% (*n* = 9) exceeded this threshold. For IL-8, 60% of participants (*n* = 18) had levels below 20 pg/mL, while 40% (*n* = 12) had higher levels.

In terms of clinical outcomes, 18 participants reported severe pain over the past 24 h, and 23 participants over the past week. Additionally, 15 participants exhibited severe functional disability. The overall mean pain score was 7 (SD = 1.74) for the last 24 h and 7.6 (SD = 1.83) for the last week. The highest pain levels (both 24-h and 1-week NRS scores) were observed in participants with IL-6 levels above 5.9 pg/mL. The mean FIQ score was 74.05 (SD = 15.4), with the highest disability observed in the group with IL-6 levels ranging from 2 to 5.9 pg/mL (mean FIQ = 80.96).

### 3.3. Significance of Associations Between Interleukins, Pain, and Disability

Statistical analysis of the associations between IL (IL-1, IL-6, IL-8) levels and clinical parameters of pain and disability revealed significant patterns that supported the study’s initial hypothesis. Specifically, elevated levels of IL-6 and IL-8 were strongly correlated with higher pain intensity and greater functional disability, as measured by both the NRS and the FIQ. The strength of these correlations varied slightly between the two scales, with IL-6 showing a stronger association with functional disability (FIQ), while IL-8 was more closely linked to pain intensity (NRS). These findings are further detailed in Section 3.3.1 and Section 3.3.2, where both NRS and FIQ scores are discussed in relation to IL levels.

#### 3.3.1. IL-6 and Functional Disability

IL-6 levels showed a significant positive correlation with FIQ scores (r = 0.65, *p* < 0.01). Participants with IL-6 levels above 2 pg/mL had significantly higher FIQ scores (mean = 80.96, SD = 10.16) compared to those with levels below 2 pg/mL (mean = 71.09, SD = 16.53), with the difference being statistically significant (*p* < 0.05). This relationship suggests that systemic inflammation mediated by IL-6 may play a crucial role in exacerbating functional disability in FM patients.

#### 3.3.2. IL-8 and Pain Perception

A positive correlation was observed between IL-8 levels and NRS scores over both the past 24 h and the past week (NRS 24 h: r = 0.58, *p* > 0.05; NRS 1 w: r = 0.62, *p* > 0.05), although these correlations were not statistically significant based on Table 2. Participants with IL-8 levels above 20 pg/mL reported higher pain intensity (NRS 24 h mean = 7.50, SD = 1.62; NRS 1 w mean = 7.75, SD = 2.18) compared to those with lower levels. However, these differences did not reach statistical significance. These findings suggest that, while IL-8 may play a role in pain modulation in FM, further investigation is needed to clarify its involvement in central sensitization and the perpetuation of chronic pain.

#### 3.3.3. IL-1 and Clinical Symptoms

Although IL-1 tended to correlate with higher pain and disability scores, the high variability among participants (SD = 57.44) limited the statistical significance of the comparisons. However, participants with elevated IL-1 levels (>13.6 pg/mL) reported higher FIQ scores (mean = 72.56, SD = 16.45) and NRS scores for 24 h (mean = 7.00, SD = 1.66) and 1 week (mean = 7.41, SD = 1.76). While these results suggest a potential involvement of IL-1 in the severity of FM symptoms, the variability (*p*-value < 0.1) implies a tendency rather than a definitive correlation.

### 3.4. Multivariate Analysis

Multiple regression analysis demonstrated that IL-6 had the strongest influence on the FIQ score, reflecting its significant contribution to disability severity (β = 0.42, *p* < 0.01). On the other hand, IL-8 was the most significant predictor for the NRS score for pain over the past week (β = 0.38, *p* < 0.05). These findings highlight that distinct interleukins are involved in different aspects of FM symptomatology, supporting the idea of targeting IL-6 in treatments aimed at reducing disability (FIQ) and focusing on IL-8 to alleviate pain (NRS).

## 4. Discussion

The aim of this study was to explore the potential association between neuroinflammation and the clinical status of FM patients.

Significant correlations were found between IL levels and clinical outcomes. In particular, elevated IL-6 levels were associated with higher FIQ scores, indicating worse disability. Another study suggested that IL-6 could modify clinical symptoms in FM patients [36] and “be responsible for the increase in the pain sensation and pain severity in FM patients” [37]. Despite some controversy, IL-6 has been suggested to play a pronociceptive role, by suppressing inhibitory neurotransmission (namely, gamma-aminobutyric acid- and glycine-induced currents), promoting cAMP response element-binding protein phosphorylation and inducing heat hyperalgesia [38]. This would lead to a feed-forward cycle of nociceptive signaling and central sensitization. It should also be noted that Hernandez et al. highlighted a significant increase in IL-6 levels in FM patients compared to healthy volunteers and that this change was independent of BMI [36,39].

Participants with higher IL-8 levels tended to report greater pain intensity and more severe functional impairment. These findings suggest that systemic neuroinflammation, as reflected by elevated IL levels, is closely associated with worse clinical outcomes in FM patients, supporting the hypothesis of a link between neuroinflammatory processes and symptom severity in this condition. Other studies have also suggested that widespread chronic pain and hypersensitivity may be mainly related to IL-1, 6, and 8 [40]. This research contributes to the growing body of evidence highlighting the pivotal role of neuroinflammation in the pathophysiology and severity of FM. Our findings, which demonstrate significant correlations between IL-6 and IL-8 levels and clinical deterioration in FM patients, are in line with earlier studies that emphasize the importance of interleukins in chronic pain modulation. In this context, it is noteworthy that recent research on osteoarthritis (OA) has also underscored the relevance of central sensitization in pain amplification. Central sensitization, a process involving the dysfunction of descending pain inhibition and the potentiation of nociceptive signals, has been implicated in the persistence of pain in chronic conditions, such as OA, and may play a similar role in FM. This suggests that, akin to OA, FM could benefit from therapeutic approaches targeting both peripheral and central mechanisms, as central pain modulation is key to maintaining and exacerbating hypersensitivity. These parallels emphasize the need for further investigation into the roles of neuroinflammation and central pain mechanisms in FM, which could lead to the development of personalized therapeutic strategies for managing this complex condition [41]. Recent evidence has suggested that the gut microbiome may play a significant role in modulating systemic inflammation and pain perception, particularly in chronic conditions such as OA. Our findings are in line with the systematic review by Herrera et al., who highlighted IL-8 as one of the most frequently identified in FM patients, primarily in relation to the pain they experience [40]. Notably, IL-8 was found to be elevated in the cerebrospinal fluid of chronic low back pain patients with intervertebral disc degeneration compared to pain-free subjects with or without disc degeneration [42]. Furthermore, chronic inhibition of the IL-8 receptors CXCR1/2 with reparixin attenuated behavioral signs of axial discomfort and radiating leg pain in SPARC-null mice. Although the underlying mechanisms remain to be elucidated, the IL-8 pathway has been suggested as a therapeutic target to reduce chronic pain.

IL-1 also showed a trend towards correlation with pain and disability, but the high variability among participants limited the statistical power. There is inconsistent evidence regarding IL-1 levels in FM patients compared to healthy controls [39,43]; nevertheless, a clear association between IL-1 and pain intensity has been shown in other chronic pain conditions, such as chronic low back pain [44]. In terms of potential mechanisms, IL-1 has been shown to regulate both excitatory and inhibitory neurotransmission [38]. In particular, similar to IL-6, IL-1β may exert a strong influence on inhibitory neurotransmission by suppressing spontaneous inhibitory postsynaptic currents and gamma-aminobutyric acid- and glycine-induced currents. However, further research involving real chronic pain patients is needed to discern the role of IL-1 in this population.

Deep research is needed to establish interleukins as a reliable biomarker in the presence of fibromyalgia or chronic pain patients. Other biomarkers could also be used in future studies to provide a more comprehensive evaluation of the inflammatory processes. The postprandial glucose response or serum levels of C-reactive protein or the lipid profile have been described as potential biomarkers in chronic pain conditions [21,45].

### 4.1. Implications for Clinical Practice

Our findings emphasize the significant role of neuroinflammation, particularly the involvement of interleukins (IL-6 and IL-8), in the clinical manifestation of FM. Monitoring IL levels may serve as a valuable tool for assessing disease severity and guiding treatment plans. Clinical settings should consider incorporating regular IL level assessments to better tailor interventions for FM patients, especially as these markers are closely associated with pain intensity and disability.

The strong association between IL-6 and IL-8 and clinical outcomes in FM patients suggests that anti-inflammatory therapies targeting these cytokines may provide a promising therapeutic avenue. We recommend that healthcare providers explore anti-inflammatory treatments as part of a comprehensive care plan for FM, especially for patients with elevated IL levels and severe symptoms.

Given the systemic nature of FM and its potential links to central sensitization, multidisciplinary treatment approaches involving rheumatologists, pain specialists, and physical therapists are critical for effective management. Rehabilitation programs should focus not only on musculoskeletal pain relief but also on reducing neuroinflammation, which may improve overall functionality and quality of life in FM patients [46].

Furthermore, emerging evidence on the role of the gut microbiome in modulating systemic inflammation suggests that dietary interventions could complement existing treatments. Clinicians may want to explore the possibility of using probiotics or other dietary modifications to mitigate gut dysbiosis and its potential contribution to FM symptoms [47].

Finally, personalized treatment strategies that account for individual cytokine profiles could enhance patient outcomes. As research progresses, IL profiling might be used for more accurate diagnosis, risk stratification, and targeted therapy, helping to optimize treatment efficacy and improve patient quality of life in FM.

### 4.2. Limitations of the Study

This study presents several limitations that must be acknowledged when interpreting the results. First, a comparison with healthy control subjects was not performed, which limits the scope of the findings regarding the role of inflammatory markers in FM patients. Other studies suggest differences in IL levels, particularly IL-6 and IL-8, between FM patients and healthy controls, which we were unable to replicate due to the lack of a control group.

Second, we did not evaluate other relevant cytokines such as IL-10, which has shown conflicting results in the literature concerning its role in FM. Without assessing a broader range of inflammatory markers, our understanding of the inflammatory landscape in FM remains incomplete.

Additionally, the medications taken by participants, such as pregabalin, which is known to influence IL levels, were not fully accounted for in this study. Moreover, we did not gather comprehensive data on other lifestyle factors such as diet and exercise, which could have affected the cytokine profiles observed [48].

Finally, the sample size of 30 patients, all of whom were female, is relatively small, limiting the generalizability of the findings. This small sample size makes it difficult to extrapolate the results to the broader FM population, including male patients.

These limitations highlight the need for future research with larger and more diverse cohorts, the inclusion of healthy control groups, and the evaluation of additional inflammatory markers to better understand the role of neuroinflammation in FM.

## 5. Conclusions

These findings highlight the importance of ILs, particularly IL-6 and IL-8, as potential biomarkers and mediators of pain severity and disability in FM patients. Further investigation, possibly using animal models, is warranted to elucidate the mechanistic pathways linking ILs, systemic inflammation and pain modulation, to potentially guide targeted therapeutic strategies for managing this complex syndrome. The identification of these associations underscores the need for continued research into the role of neuroinflammation in FM, with the potential to ensure a more accurate and early diagnosis.

## Figures and Tables

**Figure 1 cells-13-01719-f001:**
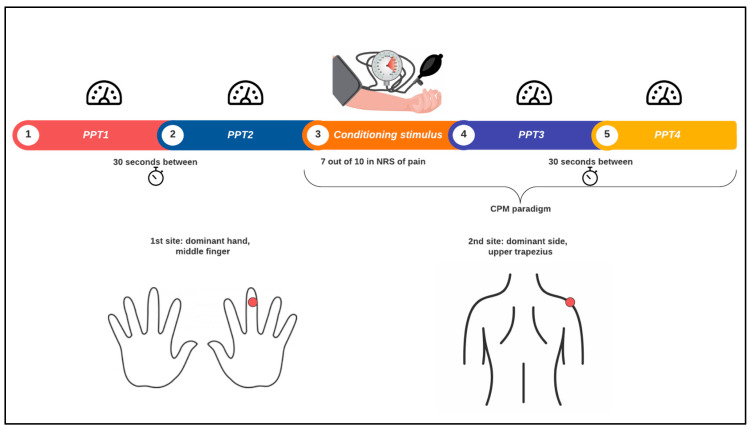
Experimental pain measurement procedure. PPT: pain pressure threshold; CPM: conditioned pain modulation; NRS: numerical rating scale.

**Table 1 cells-13-01719-t001:** Demographic, anthropometric, and clinical characteristics of the study population.

	Mean	Standard Deviation (SD)
Age (years)	50.5	8.51
Height (cm)	162.4	6.44
Weight (kg)	72.8	14.19
BMI	27.7	5.88
Years with FM	10	8.70
NRS 24 h	7	1.74
NRS 1 w	7.6	1.83
FIQ	74.1	15.43
IL-1 (pg/mL)	14.6	57.44
IL-6 (pg/mL)	1.6	0.96
IL-8 (pg/mL)	20.1	9.98

BMI: body mass index; FIQ: Fibromyalgia Impact Questionnaire, FM: fibromyalgia; IL: interleukin; NRS 24 h: numerical rating scale over the last 24 h; NRS 1 w: numerical rating scale over the last week.

**Table 2 cells-13-01719-t002:** Pain and disability in participants stratified by interleukin levels (*n* = 30).

	IL-1			IL-6		IL-8		Total (*n* = 30)
	>13.6 pg/mL	13.6–0.4 pg/mL	<0.4 pg/mL	2–5.9 pg/mL	<2 pg/mL	20–50 pg/mL	<20 pg/mL	
NRS 24 h								
Severe	1	4	13	6	12	7	11	18
M–M	2	1	9	3	9	5	7	12
Mean	7.00	7.60	6	7.22	6.90	6.92	7.06	7
SD	1.66	1.34	3.00	1.64	1.81	2.02	1.59	1.74
NRS 1 w								
Severe	2	5	16	8	15	9	14	23
M–M	1	0	6	1	6	3	4	7
Mean	7.41	8.60	7.33	8.22	7.33	7.75	7.50	7.60
SD	1.76	1.34	3.06	1.99	1.74	2.18	1.62	1.83
FIQ								
Severe	1	4	10	7	8	6	9	15
M–M	2	1	12	2	13	6	9	15
Mean	72.56	78.67	77.29	80.96	71.09	75.06	73.38	74.05
SD	16.45	14.02	11.24	10.16	16.53	14.74	16.26	15.4

IL: interleukin; NRS 24 h: numerical rating scale over the last 24 h; NRS 1 w: numerical rating scale over the last week; FIQ: Fibromyalgia Impact Questionnaire; Severe: >7 out of 10 in NRS or >59 in the FIQ; M–M: mild–moderate, <7 out of 10 in NRS or <59 in the FIQ; SD: standard deviation.

## Data Availability

The original contributions presented in the study are included in the article, further inquiries can be directed to the corresponding authors.

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
