# Peer review of "Association Between Systemic Neuroinflammation, Pain Perception and Clinical Status in Fibromyalgia Patients: Cross-Sectional Study"

_cells, 2024, doi:10.3390/cells13201719_

Round 1
Reviewer 1 Report
Comments and Suggestions for Authors
The manuscript, titled "Association between systemic neuroinflammation, pain perception and clinical status in fibromyalgia patients: cross-sectional study" , is well written and aimed to demonstrate in a small sample of patients with FM the relationship between the levels of some proinflammatory interleukins and the severity of pain.
The experimental design of this study is its strength; its weaknesses are the very limited number of patient samples and the lack of a control group.
However, this weakness is well discussed in the limits section of the study.
The only suggestion I have for the authors is that, if possible, they add information on the glucose levels of the patients evaluated to further enrich the effect of glycemia in these patients and support the information that the authors gave in the introduction.
After reviewing this small suggestion, this work is suitable for publication in this journal.
Thank you
Author Response
October 10, 2024
Dear Editorial Office,
We are pleased to submit our point-by-point response to the changes requested on our paper “Association between systemic neuroinflammation, pain perception and clinical status in fibromyalgia patients: cross-sectional study ”. We believe that we have made all the changes requested before publication and our manuscript is now responsive to all comments and suggestions. We highlighted the changes in the text with blue color to show reviewers and the editor where the changes have been made. We look forward to hearing your response and thank you for your consideration in bringing this manuscript closer to publication.
Sincerely yours,
The authors
Reviewer 1
Comments and Suggestions for Authors
The manuscript, titled "Association between systemic neuroinflammation, pain perception and clinical status in fibromyalgia patients: cross-sectional study" , is well written and aimed to demonstrate in a small sample of patients with FM the relationship between the levels of some proinflammatory interleukins and the severity of pain.
The experimental design of this study is its strength; its weaknesses are the very limited number of patient samples and the lack of a control group.
However, this weakness is well discussed in the limits section of the study.
The only suggestion I have for the authors is that, if possible, they add information on the glucose levels of the patients evaluated to further enrich the effect of glycemia in these patients and support the information that the authors gave in the introduction.
After reviewing this small suggestion, this work is suitable for publication in this journal.
Thank you
Response: Thank you very much for your nice feedback on our article. We would like to explain that we couldn’t evaluate the glycemia levels in this study. However, it is a great idea for future research.
Reviewer 2 Report
Comments and Suggestions for Authors
I attach my comments in a word document because I find that an easier program in which to to edit. Pleas excuse the formatting errors that occurred as a result of converting the PDF into a Word file.
I think this was an interesting and very worthwhile study to do as possible causes of and contributions to FM have been elusive to investigators and physicians. I think that the study was well thought out and the results are thought provoking.
As outlined in my comments in the manuscript, I have two main suggestions that if implemented I think would greatly improve the paper.
One is more general overall comment and one more specific.
The more general comment is that I found the methods section to be generally confusing and hard to follow. I like to see Methods sections that are written in such a way that would allow another researcher to repeat your study using only the information provided in the manuscript. I think that the way that the methods are written now, many questions would arise and clarifications would be needed in order to replicate what you have done. So, I suggest reworking the methods to make them more transparent and easily understandable to the reader.
The more specific recommendation is to remove the all of the references you make to the gut-brain connection from the paper. Although an interesting topic, it does not seem related to the specific work that you have done in this study and you do not explain the relevance that it has to this research. It therefore seems out of place and not connected to the work that you have done.
All in all, this is a nice piece of work and with some edits will be a really nice paper.

The quality of the English language in this manuscript was very good. The parts that were confusing were not because of the English but rather because concepts were not adequately explained.
Author Response
October 10, 2024
Dear Editorial Office,
We are pleased to submit our point-by-point response to the changes requested on our paper “Association between systemic neuroinflammation, pain perception and clinical status in fibromyalgia patients: cross-sectional study ”. We believe that we have made all the changes requested before publication and our manuscript is now responsive to all comments and suggestions. We highlighted the changes in the text with blue color to show reviewers and the editor where the changes have been made. We look forward to hearing your response and thank you for your consideration in bringing this manuscript closer to publication.
Sincerely yours,
The authors
Reviewer 2
Comments and Suggestions for Authors
I attach my comments in a word document because I find that an easier program in which to edit. Please excuse the formatting errors that occurred as a result of converting the PDF into a Word file.
Response: First of all, we would like to thank you for taking the time to make the changes to the article and for sending us that document. We have incorporated the proposed changes and would like to address some questions that are found within the document itself.
- “Why did you not also measure IL10 levels?” We appreciate your suggestion; however, due to economic reasons and limitations of the laboratory, we had to select which types of interleukins we wanted to measure based on scientific literature. Thus, we chose the 3 that we have described in the article.
- “Were they asked about all medication use or only about the medicine that they took for their FM? It seems like it was just to FM but this should be clarified” Patients with fibromyalgia have a very complex medication profile. They are usually on multiple medications and often have more than one condition in addition to fibromyalgia (migraines, irritable bowel syndrome, chronic low back pain). In our case, we specifically asked about fibromyalgia medication; however, even for the patients themselves, it is difficult to define which medication is for which symptom.
I think this was an interesting and very worthwhile study to do as possible causes of and contributions to FM have been elusive to investigators and physicians. I think that the study was well thought out and the results are thought provoking.
Response:Thank you very much for your feedback, we really appreciate it.
As outlined in my comments in the manuscript, I have two main suggestions that if implemented I think would greatly improve the paper.
One is more general overall comment and one more specific.
The more general comment is that I found the methods section to be generally confusing and hard to follow. I like to see Methods sections that are written in such a way that would allow another researcher to repeat your study using only the information provided in the manuscript. I think that the way that the methods are written now, many questions would arise and clarifications would be needed in order to replicate what you have done. So, I suggest reworking the methods to make them more transparent and easily understandable to the reader.
Response: Thank you for your feedback in this section. We changes some phrases and add extra information in order to clarify this section.
The more specific recommendation is to remove the all of the references you make to the gut-brain connection from the paper. Although an interesting topic, it does not seem related to the specific work that you have done in this study and you do not explain the relevance that it has to this research. It therefore seems out of place and not connected to the work that you have done.
Response: Thank you for your comment. We have removed the suggested section in the document according to your recommendation.
Reviewer 3 Report
Comments and Suggestions for Authors
In this manuscript, the authors observed IL-6 and IL-8 may serve as potential biomarkers and mediators for pain severity and disability in a female fibromyalgia population. Elevated IL-8 levels, for example, are correlated with increased pain perception and higher disability scores.
Comments:
1. Line 118, please clarify "see paragraph...", what does this mean?
2. Section 2.4.1, the authors should provide the detection limits for IL-1, IL-6 and IL-8.
3. Line 190-97, what is the normal range of these cytokines in their population? In addition, is C-reactive protein available as a surrogate marker for inflammation? As the study population has the prevalence of overweight, are the lipid profiles available and any correlation with IL-6/pain scores?
4. Table 1 - please provide units for IL-1, IL-6 and IL-8 (ie, pg/ml). In addition, can the authors confirm IL-8 is in pg/ml and not ng/ml?
5. Data from Section 3.4 multivariate analysis should incorporate into Table 2. Formatting in general, line 223-235 should appear below Table 2 as subtext.
6. Discussion should be more concise, Line 291-331 is rather speculative and not supported by data in this study, ie, is it relevant?
Author Response
October 10, 2024
Dear Editorial Office,
We are pleased to submit our point-by-point response to the changes requested on our paper “Association between systemic neuroinflammation, pain perception and clinical status in fibromyalgia patients: cross-sectional study ”. We believe that we have made all the changes requested before publication and our manuscript is now responsive to all comments and suggestions. We highlighted the changes in the text with blue color to show reviewers and the editor where the changes have been made. We look forward to hearing your response and thank you for your consideration in bringing this manuscript closer to publication.
Sincerely yours,
The authors
Reviewer 3
Comments and Suggestions for Authors
In this manuscript, the authors observed IL-6 and IL-8 may serve as potential biomarkers and mediators for pain severity and disability in a female fibromyalgia population. Elevated IL-8 levels, for example, are correlated with increased pain perception and higher disability scores.
Comments:
1. Line 118, please clarify "see paragraph...", what does this mean?
Response: Thank you very much for highlighting this mistake, we have already deleted it.
- Section 2.4.1, the authors should provide the detection limits for IL-1, IL-6 and IL-8.
Response: thank you for your comment. You can find this information in section 2.5 “Data Analysis”, where the participants were stratified into subgroups according to their IL levels.
- Line 190-97, what is the normal range of these cytokines in their population? In addition, is C-reactive protein available as a surrogate marker for inflammation? As the study population has the prevalence of overweight, are the lipid profiles available and any correlation with IL-6/pain scores?
Response: Thank you very much for your suggestion. We do not have data on the lipid profile of the patients participating in the database; however, we agree that it would be a very interesting question for future research.
- Table 1 - please provide units for IL-1, IL-6 and IL-8 (ie, pg/ml). In addition, can the authors confirm IL-8 is in pg/ml and not ng/ml?
Response: Thank you for your feedback. We have added this information. We confirm with the laboratory that all the IL are measured in pg/mL.
- Data from Section 3.4 multivariate analysis should incorporate into Table 2. Formatting in general, line 223-235 should appear below Table 2 as subtext.
Response: Thank you for your constructive comment. We have changed the position of the table so that its title and caption with the explanation are visible.
- Discussion should be more concise, Line 291-331 is rather speculative and not supported by data in this study, ie, is it relevant?
Response: Thank you for your feedback. We have deleted some extra information in order to make it more concise.
Round 2
Reviewer 3 Report
Comments and Suggestions for Authors
Comment #2 – Detection limit refers to sensitivity of the assays, has nothing to do with "stratified your patients into subgroups".
Comment #3 – this reviewer assume C-reactive protein is not able? The lack of CRP and lipid profiles should be included in the discussion.
Author Response
Reviewer 3
Comments and Suggestions for Authors
Comment #2 – Detection limit refers to sensitivity of the assays, has nothing to do with "stratified your patients into subgroups".
Response: Thank you for your comment. To clarify, the stratification of patients was based on the quantified interleukin concentrations (IL-1, IL-6, IL-8) obtained from the assays, and not the detection limit, which refers to the assay’s sensitivity to detect minimal concentrations. The detection limit was not relevant to the stratification process. Instead, the clinical outcomes were analyzed in relation to the stratified interleukin levels, with stratification determined according to laboratory-validated thresholds to reflect the distinct inflammatory profiles for further correlation with clinical parameters.
Comment #3 – this reviewer assume C-reactive protein is not able? The lack of CRP and lipid profiles should be included in the discussion.
Response: We appreciate your valuable suggestion regarding the inclusion of CRP and lipid profiles. In this study, we chose not to include these markers, as our primary focus was on the association between specific interleukins (IL-1, IL-6, IL-8) and clinical outcomes in fibromyalgia patients. However, we acknowledge the relevance of CRP and lipid profiles in inflammation and will address their potential value in the discussion, recommending that future studies incorporate these biomarkers to provide a more comprehensive evaluation of the inflammatory processes in fibromyalgia.